# Corrosion Studies of Temperature-Resistant Zinc Alloy Sacrificial Anodes and Casing Pipe at Different Temperatures

**DOI:** 10.3390/ma16227120

**Published:** 2023-11-10

**Authors:** Mifeng Zhao, Shaobo Feng, Fangting Hu, Hailong Geng, Xuanpeng Li, Yan Long, Wenhao Feng, Zihan Chen

**Affiliations:** 1Oil and Gas Engineering Research Institute, Tarim Oilfield Company of CNPC, Korla 841000, China; zhaomf-tlm@petrochina.com.cn (M.Z.); fengsb-tlm@petrochina.com.cn (S.F.); huft-tlm@petrochina.com.cn (F.H.); genghailong-tlm@petrochina.com.cn (H.G.); 2State Key Laboratory of Performance and Structural Safety of Petroleum Tubular Goods and Equipment Materials, CNPC Tubular Goods Research Institute, Xi’an 710077, China; lixuanpeng127@cnpc.com.cn (X.L.); longyan01@cnpc.com.cn (Y.L.); 3Shaanxi Jiuzhou Petroleum Engineering Technical Service Co., Ltd., Xi’an 710075, China

**Keywords:** high temperature resistant, zinc alloy, corrosion, SEM, sacrificial anode, polarization curve

## Abstract

In order to solve the problem of external corrosion of deep well casing in oil and gas fields, a new type of high-temperature-resistant zinc alloy sacrificial anode material was used. The temperature and corrosion resistance of the new anode material and TP140 casing were investigated by simulating the high-temperature working conditions of a deep well in an oil field using high-temperature and high-pressure corrosion tests and electrochemical tests. The results showed that at 100–120 °C, the corrosion rate of TP140 protected by a sacrificial anode was only one-tenth of that under unprotected conditions, and the minimum corrosion rate of TP140 protected by a sacrificial anode at 100 °C was 0.0089 mm/a. The results of the dynamic potential polarization curve showed that the corresponding corrosion current density of TP140 first increased and then decreased with the increase in temperature. The self-corrosion potential in sacrificial anode materials first increased and then decreased with the increase in temperature, and the potential difference with TP140 gradually decreased.

## 1. Introduction

The oil and gas reservoir in an oilfield is buried to a depth of 4500 m, with a formation pressure of 70 MPa, a temperature of 120 °C, and a complex corrosive environment. Since 1998, some development wells have simplified their well structure, and large sections of production casing have been produced with discontinuous cementing cement or uncured sections, resulting in very serious casing losses. The research and analysis concluded that casing corrosion in oil wells is mainly due to external corrosion, while CO_2_, Cl^−^, and highly mineralized formation water are important causes of casing loss under high temperature and pressure conditions.

Sacrificial anode cathodic protection is recognized as an effective technique for controlling external corrosion, but the effectiveness of the protection is directly related to the performance of the sacrificial anode material itself [1,2,3]. Zinc alloy is one of the most commonly used anode materials, with abundant resources, a simple manufacturing process, excellent electrochemical properties, etc. [4]. In order to improve the electrochemical performance of sacrificial anode materials, scholars have continuously added various elements to improve the electrochemical performance of aluminum-zinc alloys. S.M.A. Shibli et al. [5] added 0.2% nanocerium oxide particles to the anode matrix Al + 5% Zn alloy, which significantly improved the metallurgical properties of the anodes and increased their efficiency from 44.43% to 78.62%. The effect of lanthanum on the microstructure and electrochemical properties of Al-Zn-based sacrificial anode alloys was investigated by Ma et al. [6] It was found that the addition of La improved the electrochemical properties of the Al-5Zn-0.03In-1Mg-0.05Ti-0.5La (wt.%) alloy and ameliorated the non-uniform corrosion of Al_2_LaZn_2_ precipitated particles. Luo et al. [7] designed and prepared Al-Zn-0.03 In-1.30 Mg sacrificial anode materials with different Zn contents. As the Zn content increased, the Al-Zn-0.03In-1.30Mg anode grain became finer, and the metallographic structure became more uniform. The 0.60% Zn content of the anode could significantly reduce the heavy metal load of Zn in the sacrificial anode materials in the marine environment.

Moreover, zinc alloy has a low melting point, good flowability, easy fusion welding, and corrosion resistance in the atmosphere [8,9,10]. However, in the use of zinc-based sacrificial anode materials, the environmental medium and temperature have a significant effect on their electrochemical performance [11,12,13]. For example, in a high-temperature seawater environment above 60 °C, the potential reversal of the zinc anode relative to the protected steel material will occur, and the steel material will no longer be protected but will accelerate corrosion [14]. Therefore, the application of general zinc alloys in high-temperature environments is limited, and there is a need to investigate new high-temperature-resistant anode materials.

In this work, based on the Zn-Al-Cd alloy provided by GB/T4950-2002 [15], a new type of zinc alloy sacrificial anode material is formed by strictly controlling the content of Al elements and adding Mn and Mg, etc. The temperature resistance and corrosion resistance of the new anode material are investigated by simulating the high-temperature working conditions of a deep well in an oil field.

## 2. Experimental

### 2.1. Materials

High-temperature and high-pressure corrosion test materials were taken from the casing with zinc alloy sacrificial anode sleeve casing (as shown in Figure 1) and TP140 casing; the shape and dimensions of the samples are shown in Figure 2. The composition of TP140 is given in Table 1. Sacrificial anode material is melted by adding the Re element to Zn-Al-Cd alloy to obtain: Al–5.5000, Zn–0.0300, In–1.5000, Mg–0.0500, Ti–0.0167, where the Ti/Fe mass ratio is 3 to 5 and Cd is 0.05% to 0.12%.

Before the test, the samples were sanded in turn with 240#, 400#, 600#, 800#, and 1000# SiC paper [16], then cleaned with acetone, degreased, dried with cold air, and weighed. The actual size of the samples was measured using vernier calipers, and the weight and dimensions of each sample were recorded.

### 2.2. Corrosion Experiments

High temperature and pressure corrosion and galvanic corrosion tests were carried out in the autoclave (Figure 3); the specific test parameters and conditions are shown in Table 2. For the galvanic corrosion test, bolts were used to connect the zinc alloy sacrificial anode specimen to the TP140 specimen, using bolts to establish direct contact between the two samples, thus creating a galvanic couple pair to simulate the anodic protection of the casing by the sacrificial anode sleeve, and then the galvanically connected samples were mounted in a special hanger for the autoclave. The electrochemical tests were carried out using a three-electrode cell with a working electrode (CORR T350-P3000, Long Beach, CA, USA), a platinum counter electrode (CORR T350-P3000, Long Beach, CA, USA), and an external pressure-balanced Ag/AgCl reference electrode. The reference electrode was housed in a separate compartment that was maintained at ambient temperature and system pressure via a solution bridge.

For the high temperature and pressure corrosion test, the unconnected zinc alloy sacrificial anode sample and the TP140 sample were mounted directly in the special hangers of the autoclave with a certain distance between the two samples to ensure no contact between them. The simulated solution prepared with analytical reagent and deionized water was introduced into the autoclave, and the autoclave lid was sealed. High-purity N_2_ was used to deoxygenate the solution in the autoclave at a flow rate of not less than 100 mL/min with a solution deoxygenation time of not less than 1 h [17]. Then, the autoclave was heated to the test temperature specified in Table 1, and after the temperature stabilized, 2 MPa of high-purity CO_2_ was injected. Finally, high-purity N_2_ was used to increase the total test pressure in the autoclave to 10 MPa, and the test began.

After the 360-h corrosion test, the samples were removed, cleaned with deionized water and alcohol, and then blown dry with cold air [18]. For both sacrificial anode-protected and unprotected TP140 samples, two of each set of three parallel samples were removed for corrosion rate analysis, and the samples were soaked in a scale remover to remove the corrosion product scale and then dried in anhydrous ethanol after scale removal. The macroscopic morphology of the treated samples was observed using an ultra-deep-field microscope. Finally, the samples were placed in a desiccator, measured for dimensions, and weighed after 1 h. For the calculation of the corrosion rate, the NACE SP0775-2018 [19] standard “Preparation and Installation of Corrosion Hangings and Analysis of Test Data in Oilfield Production” was used, and the calculation formula is as follows:(1)$$\mathrm{CR}=\frac{3.65\times 10^{5}\times W}{\mathrm{ATD}}$$

In the formula:

CR—corrosion rate, mm/a;

W—weight loss, g;

A—surface area of the metal specimen, mm^2^;

T—corrosion test time, d;

D—the density of the specimen, g/cm^3^.

The sacrificial anode specimens with TP140 attached, the sacrificial anode specimens without the casing attached, and the remaining TP140 with and without sacrificial anode protection were analyzed for corrosion products using ultra-deep field microscopy. The TESCAN VEGA II scanning electron microscope (SEM) equipped with energy dispersive spectroscopy (EDS) was used to observe the microscopic morphology and energy spectrum of the specimens.

### 2.3. Electrochemical Measurements

The electrochemical tests were carried out in a high-temperature and high-pressure electrochemical reactor using a VersaSTAT 3F electrochemical workstation with a three-electrode system. In this casing, the working electrode was a stainless steel specimen spot welded to a working electrode manufactured by Tohshin, Japan, with a 20 mm × 20 mm × 1 mm platinum electrode as the auxiliary electrode and an external Ag/AgCl electrode as the reference electrode. The Ag/AgCl electrode core of the electroactive accessory was connected to the high-temperature and high-pressure vessel by a non-isothermal salt bridge (0.1 mol/Kg KCl) in the low-temperature region. All electrochemical experiments were carried out under deoxygenated environmental conditions, using high-purity N_2_ by the bubbling method to remove oxygen from the produced aqueous solution for 4 h. Subsequently, residual oxygen was removed by CO_2_ gas prior to the experiment. Prior to all electrochemical tests, the sample surface was polarized by constant potential polarization at −1.3 V (relative to the reference electrode) for 180 s to remove oil and oxide scale from the sample surface, followed by stabilization in the test medium for 60 min. To ensure the accuracy of the experiments, all electrochemical tests were repeated three times. The high-temperature and high-pressure electrochemical reactor is equipped with a high-temperature and high-pressure reference electrode. The specific model is a UHP (TOSHIN) type external Ag/AgCl electrode. The working electrode and counter electrode are both imported UHP (TOSHIN) type electrodes. The potential of the reference electrode is converted to standard hydrogen potential (SHE) during use. The specific potential changes with temperature, as shown in Equation (2) [20]:(2)$$E_{SHE}=E_{obs}+0.2866-0.001(T-T_{0})+1.754\times 10^{-7}(T-T_{0}{)}^{2}\;-3.03\times 10^{-9}{\left(T-T_{0}\right)}^{3}$$
where *E_SHE_* is the potential relative to the standard hydrogen electrode, *E_obs_* is the potential obtained from the extreme environment measurements, *T* is the temperature of the experiment (°C), and *T*_0_ is the room temperature (25 °C). The potential scan rate of the dynamic potential polarization curve tests is 0.333 mV/s, the potential scan interval is −300 mV (relative to the OCP) to 1.6 V (relative to the reference electrode), and the test was stopped when the anode current density was greater than or equal to 10 mA/cm^2^.

## 3. Results and Discussion

### 3.1. Weight Loss Measurements

Figure 4 shows the corrosion rate of TP140 casing protected by sacrificial anodes and TP140 casing without sacrificial anode protection. As can be seen from the figure, the corresponding corrosion rates of TP140 are 0.25 mm/a, 0.29 mm/a, and 0.27 mm/a under the corrosive environments of 100 °C~120 °C and 2 MPa CO_2_, respectively, and the corrosion rate was the largest at 110 °C. The corrosion rate of TP140 protected by sacrificial anodes showed the same pattern, with a minimum corrosion rate of 0.0089 mm/a at 100 °C. According to the NACE SP0775-2018 standard, a corrosion rate below 0.025 mm/a is considered mild corrosion. However, as the temperature increased, the corresponding corrosion rate reached moderate corrosion (0.025 mm/a~0.125 mm/a). The protection efficiency of sacrificial anode materials is as follows: 96.44% at 100 °C, 88.03% at 110 °C, and 89.52% at 120 °C.

M. Ueda has studied the effect of different temperatures on the corrosion rate of Cr-bearing steels. When the temperature is between 50 and 250 °C, the maximum value of 1Cr steel appears at 110 °C. According to the CO_2_ corrosion model of Cr-bearing steel, a spontaneous passivation corrosion product including FeCO_3_ reinforced by an amorphous enriched-hydrated Cr-O compound is formed at lower temperatures. As the temperature increases, a codeposition of non-protective porous FeCO_3_ and enriched hydrated Cr-O compounds is formed. The corrosion behavior is similar to that of carbon steel at the maximum corrosion rate. Thereafter, as the temperature continues to rise, a dense and adherent protective FeCO_3_ film is formed, and the rate of corrosion reaction decreases. As the Cr content of TP140 is close to 1%, the corrosion rate of TP140 at 110 °C appears to be extremely high.

### 3.2. Corrosion Morphology Analysis

#### 3.2.1. Macroscopic Morphology

The macroscopic morphology of the protected TP140 and the sacrificial anode of the joint sleeve after corrosion at different temperatures is shown in Figure 5. At 100 °C, the surface product film of the sacrificial anode-protected TP140 sample was dense and continuous, and there were no obvious pits on the surface of the sample after the film was removed. The surface corrosion products of the sacrificial anode sample connected to the TP140 sample were relatively uniform, and there was no large-scale peeling. At 110 °C, there was a large amount of coating adhesion on the surface of the TP140 sample protected by the sacrificial anode; the surface of its substrate still showed a metallic luster; and there was no obvious distribution of corrosion pits after removing the scale. The sacrificial anode sample connected to the TP140 sample shows a uniform corrosion morphology, and a large amount of coating adhesion can be seen on the surface of the sample. The sample did not show any macroscopic fractures or damage. At 120 °C, a large amount of coating adhesion was present on the surface of the TP140 sample protected by the sacrificial anode. The surface of the substrate after removal of the film showed a metallic luster, and there was no obvious distribution of corrosion pits. The TP140 sample protected by the sacrificial anode not only appeared dark black-brown, but white crystalline particles could also be seen. Although the surface of the sample after removal of the film showed a metallic luster, a rough and small pitted morphology could be seen; the sacrificial anode sample connecting the TP140 sample showed severe damage and cracking.

The surface morphology of the samples was observed at 100 °C, 110 °C, and 120 °C using an ultradeep field microscope, as shown in Figure 6. At 100 °C, neither the sacrificial anode-protected TP140 samples nor the unprotected TP140 samples suffered severe localized corrosion, and the maximum pitting depths on their surfaces were 1.0 μm and 2.2 μm, respectively. At 110 °C, the corrosion on the surface of the substrate of the TP140 specimen protected by sacrificial anodes was slight, and the depth of corrosion pits on the specimen surface was about 1 μm. A large number of corrosion pits were visible on the surface of the substrate of the TP140 specimen not protected by sacrificial anodes, and the maximum depth of corrosion pits on the surface of the specimen exceeded 25 μm as measured by 3D morphometry. Localized corrosion pits existed on the TP140 specimen protected by sacrificial anodes at 120 °C; the specimen had localized corrosion pits, and the depth of the corrosion pits on the surface of the specimen exceeded 34 μm. The depth of the corrosion pits exceeded 6 μm, which was significantly higher than at 110 °C. The TP140 specimen protected by sacrificial anodes showed a large distribution of corrosion pits with a depth of more than 26 μm.

The surface morphology of the two sacrificial anodes at different temperatures is shown in Figure 7. At 100 °C, there was no significant corrosion cracking on the surface morphology of the two sacrificial anodes. At 110 °C, the surface morphology of the sacrificial anode connected to the TP140 casing was similar to that of the sacrificial anode not connected to the TP140 casing, with a large amount of coating adhering to the surface, whereas the coating of the sacrificial anode connected to the TP140 casing was visible as a crack feature. At 120 °C, the surface morphology of the sacrificial anode connected to the TP140 casing was similar to that of the sacrificial anode not connected to the TP140 casing, with severe surface corrosion and a large number of cracks.

#### 3.2.2. Microscopic Morphology and Phase Analysis

Figure 8a–d shows the surface micromorphology and EDS results of the TP140 casing samples protected by sacrificial anodes at 100 °C and the unprotected casing samples. Blocky crystalline products were formed on the surface of both samples, and EDS analysis of the products showed that they were mainly dominated by O, Ca, Mg, and Zn, with a significantly higher content of Ca in the products of the unprotected samples. Figure 8e–h shows the surface micromorphology and EDS results for the sacrificial anode samples with and without the TP140 casing attached, respectively. Both types of sacrificial anode specimens were covered with massive corrosion products, and the results of EDS analysis showed that the surfaces of both samples were mainly covered with O, Mg, Cl, and Zn.

Figure 9a,b shows the XRD patterns of the TP140 casing specimens protected by sacrificial anodes and the sacrificial anode specimens with the TP140 casing attached at 100 °C. As can be seen from the XRD spectrum, the main corrosion products of TP140 associated with the sacrificial anode at 100 °C are FeCO_3_, CaCO_3_, (CaMg)CO_3_, and Fe, which is mainly due to the detachment of some corrosion products from the substrate surface. On the sacrificial anode surface, in addition to the substrate element Zn, the main corrosion products are ZnO, Zn(OH)_2_, AlCl_3_, Mg(ClO_4_)_2_, and so on.

Figure 10a–d shows the surface micromorphology and EDS results of TP140 casing specimens protected by sacrificial anodes and without sacrificial anodes at 110 °C. Both types of TP140 specimens had similar surface micromorphology, with a large number of massive crystalline products. The EDS results indicated that the surface of the specimen was dominated by the elements Ca and O. Figure 10e–h shows the surface micromorphology and EDS results of the sacrificial anode specimens with TP140 casing attached and without TP140 casing attached, respectively. Both types of sacrificial anode specimens had similar surface micromorphological characteristics and were dominated by massive corrosion products. The results of the EDS analysis showed that in addition to Ca and O elements, the surface of the sacrificial anode sample without the TP140 casing specimen attached also showed a distribution of Zn elements.

Identically, microscopic morphology observation and EDS analysis were carried out on the specimens after the corrosion test under the conditions of 120 °C and 2 MPa CO_2_. Figure 11a–d shows that there are still a large number of blocky crystalline morphologies on the surface of the sacrificial anode-protected sample. In addition to the crystalline morphology, loose corrosion products with no fixed morphology could also be observed on the surface of the sample without sacrificial anode protection. The surface energy spectrum analysis results of the TP140 casing sample protected by a sacrificial anode showed that the surface of the sample was mainly distributed with Ca, O, Fe, Cl, and a small amount of Na and Mg elements. The surface energy spectrum analysis results of a TP140 casing sample without sacrificial anode protection showed that the surface of the sample was mainly distributed with O, Fe, and Cl elements. The surface microstructure of sacrificial anode samples with and without TP140 casing connections is shown in Figure 11e–h. In addition to the blocky crystalline morphology, loose needle-like or flocculent corrosion products were observed on the surfaces of both types of sacrificial anode samples. Ca, Na, O, and Fe elements were mainly distributed on the surface of both types of samples, and Zn element distribution could also be seen on the surface of the sacrificial anode sample without TP140 casing connection.

### 3.3. Electrochemical Characteristics

Figure 12 shows the results of high temperature and high pressure electrochemical tests of TP140 and sacrificial anode materials in the simulated formation water environment, and the relevant electrochemical parameters obtained from the table extrapolation are listed in Table 3. From Figure 12a, it can be seen that the anodes exhibit a characteristic of anodic dissolution without any passivation zone. As the temperature increases from 100 °C to 120 °C, the corresponding self-corrosion potentials are −394.64 mV_SHE_, −416.68 mV_SHE_, and −412.74 mV_SHE_, showing a tendency to first decrease and then increase. Meanwhile, the self-corrosion current density also shows the same pattern, which is consistent with the immersion results. The self-corrosion potential test results of the sacrificial anode in Figure 12b showed the opposite trend to the test results of TP140, which showed the trend of first increasing and then decreasing, while the corrosion current density showed the same law as the test results of TP140. In addition, the potential difference ∆E between TP140 and sacrificial anode with the increase of temperature were 407.23 mV, 159.89 mV, and 335.14 mV, indicating that with the increase of temperature, the electric coupling corrosion driving force between TP140 and sacrificial anode material first increased and then decreased, combined with the change rule of corrosion rate of TP140, which further explains the efficiency of sacrificial anode protection at 100 °C > 120 °C > 110 °C.

## 4. Conclusions

The corresponding corrosion rates of TP140 show an increasing and then decreasing trend under the corrosive environment of 100 °C~120 °C and 2 MPa CO_2_, while the corrosion rate of TP140 with sacrificial anode protection was significantly lower, which was only one tenth of that under unprotected conditions.As the temperature increases, a large number of massive corrosion products appear on the surface of the sacrificial anode material, and the depth of the pitting pits increases while the pitting tendency of the protected TP140 surface increases.The protection efficiency of the sacrificial anode decreases when the temperature exceeds 100 °C. It is recommended to use sacrificial anodes below 100 °C.

## Figures and Tables

**Figure 1 materials-16-07120-f001:**
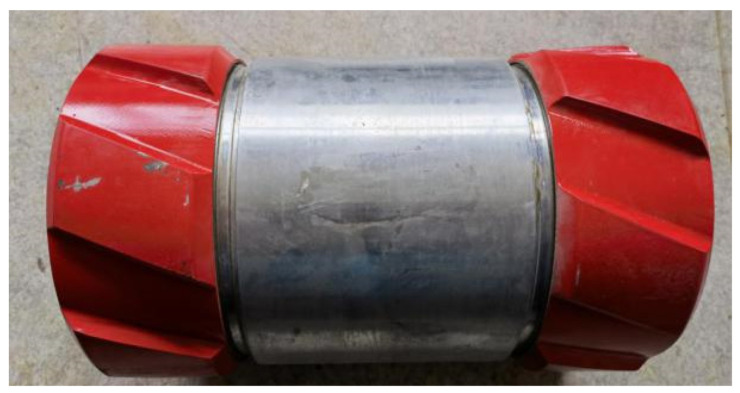
Zinc alloy sacrificial anode sleeve.

**Figure 2 materials-16-07120-f002:**
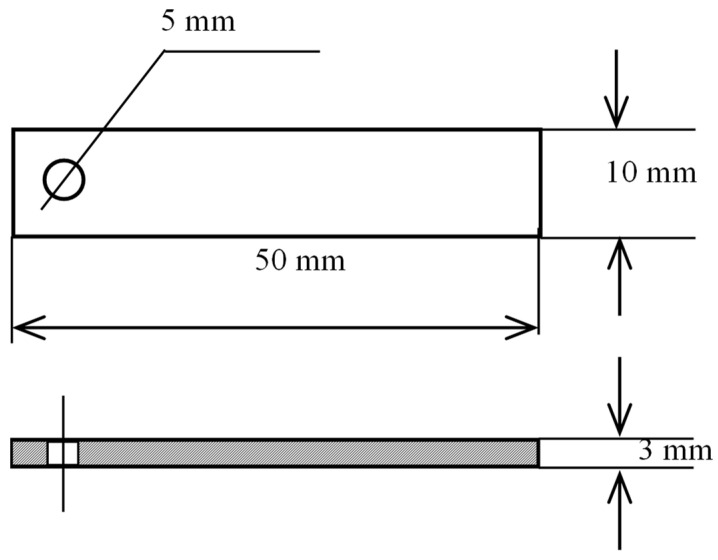
Sample size specifications for corrosion testing.

**Figure 3 materials-16-07120-f003:**
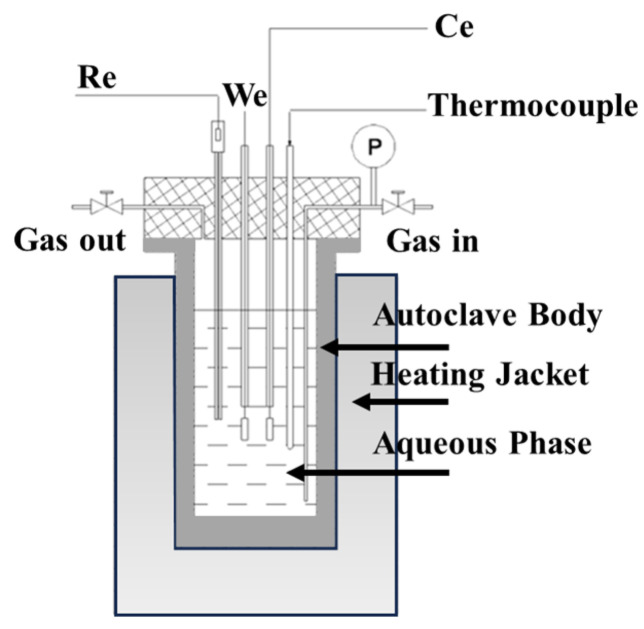
Schematic diagram of the reaction vessel test.

**Figure 4 materials-16-07120-f004:**
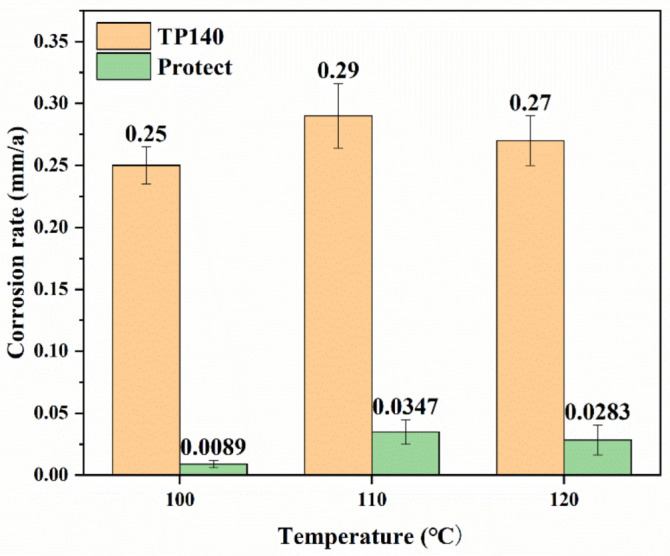
Corrosion rate of TP140 casings protected by sacrificial anodes and without sacrificial anode protection.

**Figure 5 materials-16-07120-f005:**
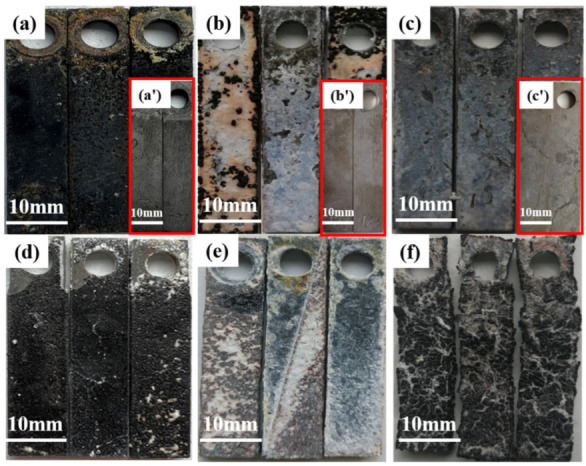
Macromorphology of specimens at two different temperatures: (**a**,**a’**,**d**) 100 °C; (**b**,**b’**,**e**) 110 °C; (**c**,**c’**,**f**) 120 °C; (**a**–**c**) TP140 casing protected by sacrificial anodes; (**a’**,**b’**,**c’**) TP140 casing protected by sacrificial anodes after scale removal; (**d**–**f**) sacrificial anode for connecting the TP140 casing.

**Figure 6 materials-16-07120-f006:**
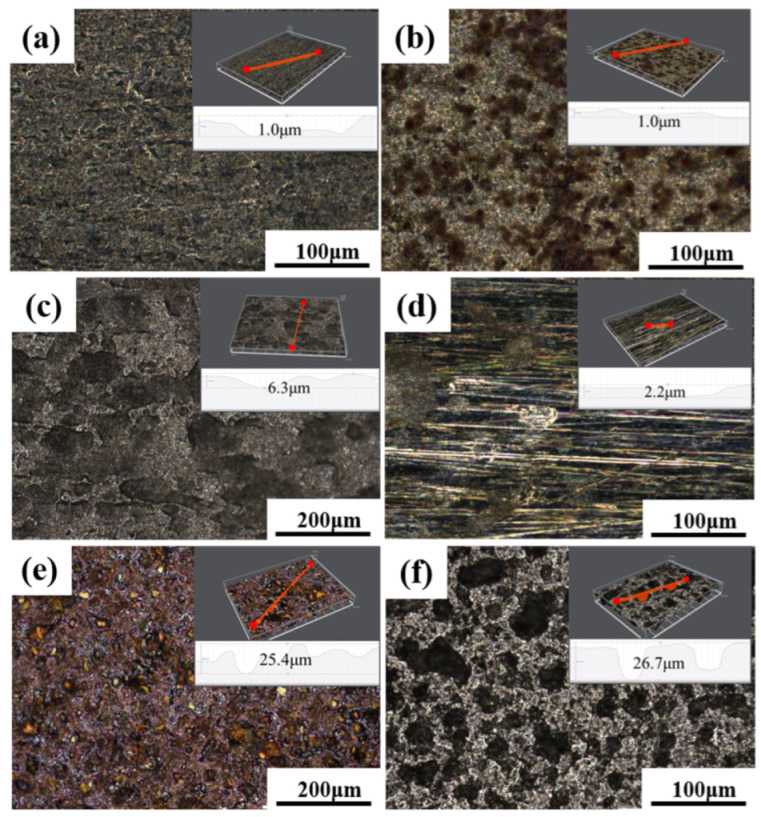
Ultra-depth-of-field profiles of specimens at two different temperatures: (**a**,**d**) 100 °C; (**b**,**e**) 110 °C; (**c**,**f**) 120 °C; (**a**–**c**) TP140 casing protected by sacrificial anodes; (**d**–**f**) TP140 casing not protected by sacrificial anodes.

**Figure 7 materials-16-07120-f007:**
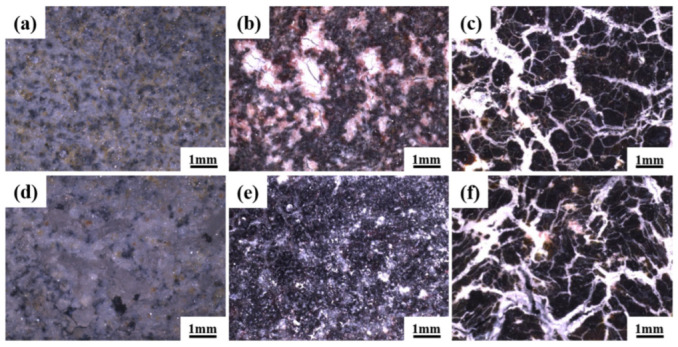
Ultra-depth-of-field profiles of sacrificial anodes at two different temperatures: (**a**,**d**) 100 °C; (**b**,**e**) 110 °C; (**c**,**f**) 120 °C; (**a**–**c**) sacrificial anode connected to TP140 casing; (**d**–**f**) sacrificial anode without TP140 casing attached.

**Figure 8 materials-16-07120-f008:**
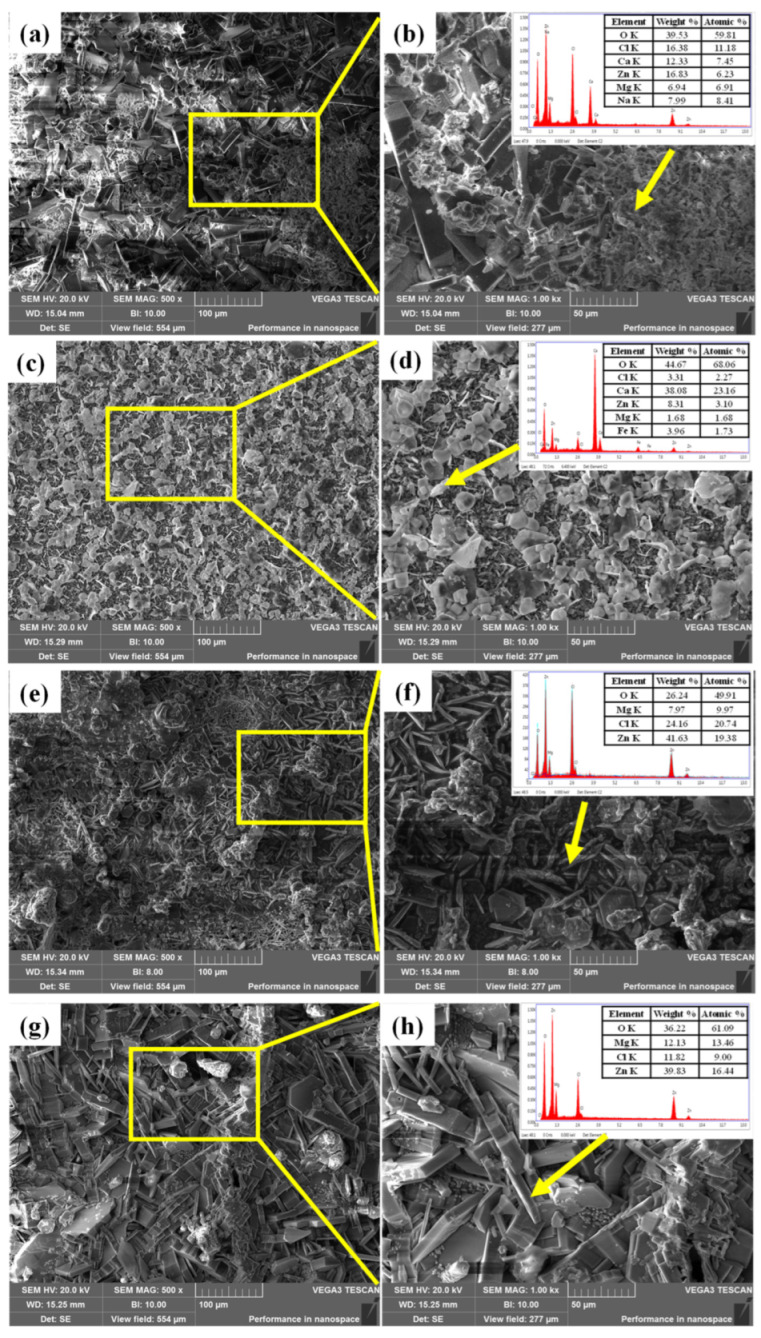
Microstructural morphology of TP140 casing and sacrificial anode samples at 100 °C: (**a**,**b**) TP140 casing samples protected by sacrificial anodes; (**c**,**d**) TP140 casing samples not protected by sacrificial anode; (**e**,**f**) sacrificial anode samples with the TP140 casing attached; (**g**,**h**) sacrificial anode samples without the TP140 casing attached.

**Figure 9 materials-16-07120-f009:**
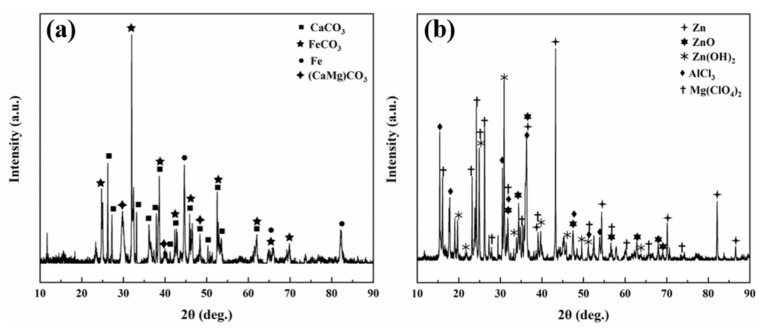
XRD patterns of the TP140 casing specimens protected by sacrificial anodes (**a**) and the sacrificial anode specimens with the TP140 casing attached (**b**) at 100 °C.

**Figure 10 materials-16-07120-f010:**
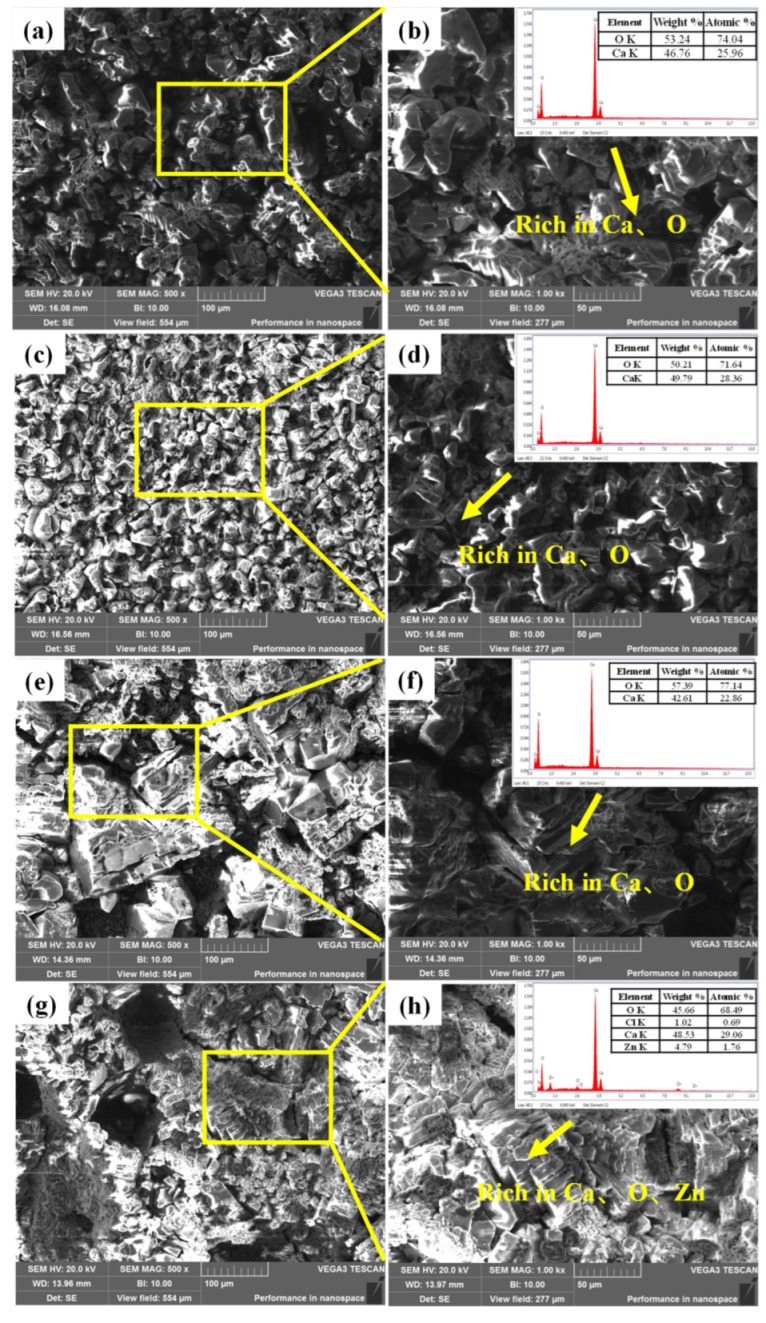
Microstructural morphology of TP140 casing and sacrificial anode samples at 110 °C. (**a**,**b**) TP140 casing samples protected by sacrificial anodes; (**c**,**d**) TP140 casing samples not pro-tected by sacrificial anode; (**e**,**f**) sacrificial anode samples with the TP140 casing attached; (**g**,**h**) sacrificial anode samples without the TP140 casing attache.

**Figure 11 materials-16-07120-f011:**
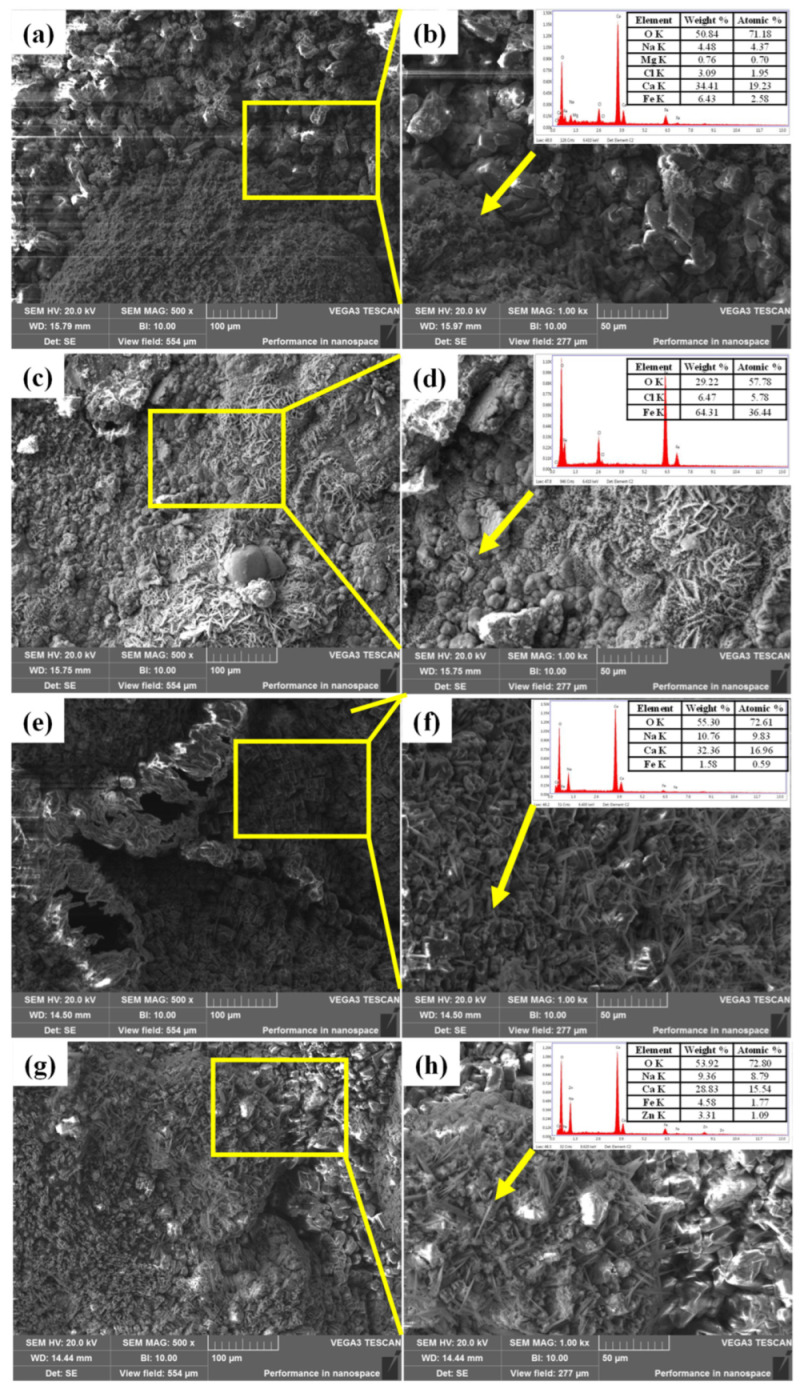
Microstructural morphology of TP140 casing and sacrificial anode samples at 120 °C. (**a**,**b**) TP140 casing samples protected by sacrificial anodes; (**c**,**d**) TP140 casing samples not pro-tected by sacrificial anode; (**e**,**f**) sacrificial anode samples with the TP140 casing attached; (**g**,**h**) sacrificial anode samples without the TP140 casing attache.

**Figure 12 materials-16-07120-f012:**
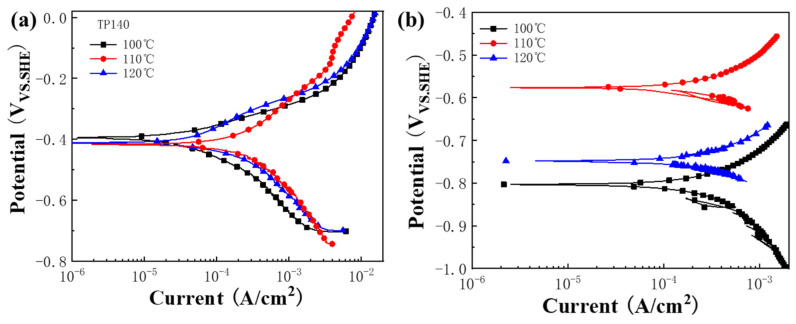
The dynamic potential polarization curves for (**a**) TP140 and (**b**) the sacrificial anode material.

**Table 1 materials-16-07120-t001:** Elemental composition of TP140 (wt.%).

C	Si	Mn	Cr	Mo	V	Fe
0.26	0.28	0.59	0.99	0.79	0.15	Bal.

**Table 2 materials-16-07120-t002:** Corrosion experimental parameters.

Temperature (°C)	100 °C, 110 °C, 120 °C
Time (h)	360
Total pressure (MPa)	10
CO_2_ partial pressure (MPa)	2
Sample	Two sets of galvanic corrosion specimens: (i) TP140 specimens protected by sacrificial anodes (3 parallel specimens); (ii) sacrificial anode specimens with TP140 attached (3 parallel specimens). Two sets of corrosion tests: (i) sacrificial anode specimens without casing attached (3 parallel specimens); (ii) TP140 specimens without sacrificial anode attached (3 parallel specimens).
Solution	K^+^: 435 mg/L; Na^+^: 55,498 mg/L; Ca^2+^: 10,000 mg/L; Mg^2+^: 1000 mg/L; SO_4_^2−^: 430 mg/L; HCO_3_^−^: 100 mg/L; Cl^−^: 100,000 mg/L.

**Table 3 materials-16-07120-t003:** Fitting electrochemical parameters.

	TP140		Sacrificial Anodes	
	E_corr_	j_corr_	E_corr_	j_corr_
100	−394.64 mV_SHE_	25.89 μA/cm^2^	−801.87 mV_SHE_	247.38 μA/cm^2^
110	−416.68 mV_SHE_	126.32 μA/cm^2^	−576.57 mV_SHE_	266.98 μA/cm^2^
120	−412.74 mV_SHE_	55.78 μA/cm^2^	−747.88 mV_SHE_	188.78 μA/cm^2^

## Data Availability

All relevant data are contained in the present manuscript.

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
