# Peer review of "Corrosion Studies of Temperature-Resistant Zinc Alloy Sacrificial Anodes and Casing Pipe at Different Temperatures"

_materials, 2023, doi:10.3390/ma16227120_

Round 1

Reviewer 1 Report

In this work, the authors have studied the corrosion resistance of zinc alloy-based sacrificial anodes and casing pipes at 100 – 120 °C. The experiments were to simulate the working conditions of a deep well in an oil field (the experiments were carried out in an autoclave at high CO2 partial pressure). The work is interesting and worthy of publication subject to revision. The following comments should be considered:

1.Specify the full chemical composition of the sacrificial anode material. It is not enough to mention that it was a Zn-Al-Cd alloy. The concentrations of the elements must be specified.

2.The chemical composition of the TP140 casing/tubing steel should also be given in the manuscript.

3.It is recommended to number all lines in the manuscript to facilitate orientation in the paper.

4.A sketch or a photo of galvanic couples studied (sacrificial anode-casing steel) should be provided in the manuscript for the sake of illustration.

5.The corrosion rate equation on p. 4 should be numbered.

6.The manuscript does not provide a reference for the NACE SP0775-2018 standard.

7.It is recommended to use the dimension mm/year instead of mm/a (a=annum) for the corrosion rate (Fig. 3).

8.The inset images in Fig. 4 are very small. It is recommended to increase their size. You should show a maximum of two images in a row.

9.The scale bars in Figs. 4 and 5 do not have any units.

10.Images in Figs. 6, 7 and 8 are again very small. The insets are hard to read. Use a maximum of 2 images in a row.

11.The paper lacks x-ray diffraction results (XRD). The phase constitution of corrosion products should be characterized by XRD. EDS analysis is not sufficient.

12.Electrochemical parameters are not sufficiently analyzed (Table 2). The entire section on p. 9 has only 7 lines. You should calculate the corrosion rate from corrosion currents using Faraday’s law and compare it with weight loss measurements (Fig. 3). Any difference should be discussed.

13.Since the authors studied the corrosion behavior at 3 different temperatures, the effect of temperature should be adequately discussed. It is recommended to estimate an activation energy of the corrosion process and compare it with previous reports.

Author Response

Manuscript with a title: “Corrosion studies of temperature-resistant zinc alloy sacrificial anodes and casing pipe at different temperatures”

Number: materials-2453392

Dear Professor:

Thank you for your letter and the reviewer's comments on our manuscript. These comments are all valuable and very helpful in revising and improving our paper, as well as providing important guidance for our research. We have carefully studied the comments and have made corrections that we hope will meet with approval. The changes are highlighted in red in the revised manuscript. The point-by-point responses to the reviewers' and editor's comments are as follows:

Reviewer #1: In this work, the authors have studied the corrosion resistance of zinc alloy-based sacrificial anodes and casing pipes at 100 – 120 °C. The experiments were to simulate the working conditions of a deep well in an oil field (the experiments were carried out in an autoclave at high CO2 partial pressure). The work is interesting and worthy of publication subject to revision. The following comments should be considered:

ResponseThank you for the reviewers' comments on the manuscript. We have corrected the errors and problems as suggested, and the changes are highlighted in red in the revised manuscript.

1) Specify the full chemical composition of the sacrificial anode material. It is not enough to mention that it was a Zn-Al-Cd alloy. The concentrations of the elements must be specified.

Response: Thank you for the review’s point of view. The chemical composition of the sacrificial anode material is given in section 2.1 of the manuscript.

2) The chemical composition of the TP140 casing/tubing steel should also be given in the manuscript.

Response: Thank you for the review’s point of view. The chemical composition of the TP140 casing/tubing is given in Table 1.
3) It is recommended to number all lines in the manuscript to facilitate orientation in the paper..

Response: Thank you for the review’s point of view.
4)
A sketch or a photo of galvanic couples studied (sacrificial anode-casing steel) should be provided in the manuscript for the sake of illustration.

Response: Thank you for the review’s point of view. We give a picture of the sacrificial anode in Figure 1 and a picture of the connected sacrificial anode and 140 casing after the test in Figure 4.

5) The corrosion rate equation on p. 4 should be numbered.

Response: Thank you for the review’s point of view. We have numbered the corrosion rate equations on page 4.

6) The manuscript does not provide a reference for the NACE SP0775-2018 standard.

Response: Thank you for the review’s point of view. We have provided the NACE SP0775-2018 standard in our references.

 7) It is recommended to use the dimension mm/year instead of mm/a (a=annum) for the corrosion rate (Fig. 3).

Response: Thank you for the review’s point of view. In the field of oil and gas field corrosion, the general expression of the corrosion rate is usually used mm/a, so we choose mm/a to express.

 8) The inset images in Fig. 4 are very small. It is recommended to increase their size. You should show a maximum of two images in a row.

Response: Thank you for the review’s point of view. We have reworked image 5 (original image 4).

9) The scale bars in Figs. 4 and 5 do not have any units.

Response: Thank you for the review’s question. The two maps have been redrawn to scale.

 10) Images in Figs. 6, 7 and 8 are again very small. The insets are hard to read. Use a maximum of 2 images in a row.

Response: Thank you for the review’s question. Three maps have been redrawn.

11) The paper lacks x-ray diffraction results (XRD). The phase constitution of corrosion products should be characterized by XRD. EDS analysis is not sufficient.

Response: Thank you for the review’s point of view. We have added XRD patterns of TP140 cased specimens protected by sacrificial anodes at 100 °C.

12) Electrochemical parameters are not sufficiently analyzed (Table 2). The entire section on p. 9 has only 7 lines. You should calculate the corrosion rate from corrosion currents using Faraday’s law and compare it with weight loss measurements (Fig. 3). Any difference should be discussed.

Response: Thank you for the review’s point of view. We have reworked the contents of section 3.3.

13) Since the authors studied the corrosion behavior at 3 different temperatures, the effect of temperature should be adequately discussed. It is recommended to estimate an activation energy of the corrosion process and compare it with previous reports.

Response: Thank you for the review’s point of view. We have calculated the protection efficiency of the sacrificial anodes at different temperatures and explained the corrosion rate of TP140 at different temperatures in section 3.1, and we have refined the interpretation of the electrochemical results in section 3.3, explaining the reason for the difference in the protection efficiency of the sacrificial anode stages at different temperatures.

Best wishes

Yours sincerely

Wenhao Feng

Aug 15, 2023

PS: If the answer was not agreeable to the reviewer, we will try our best to find the reasonable explanation again.

Reviewer 2 Report

Review of the manuscript “Corrosion studies of temperature-resistant zinc alloy sacrificial anodes and casing pipe at different temperatures”

The paper topic is actual and hot. The experimental procedures of corrosion tests being applied are reasonable. However, a description of the results looks poor. There is no X-ray data about the phase composition of corrosion products. English is not good. The paper looks like a report, not a scientific article. So, unfortunately, I cannot recommend it for publication.

Some comments are below.      

Abstract

·        The abbreviation TP140 is not clear. Please, specify what is “TP140 deep well casing”

·        English is not proper in the sentence “…The self-corrosion potential in the sacrificial anode material gradually increased with the increase of temperature, and the potential difference with TP140 gradually decreased”

·        There is no conclusion in the Abstract

1.      Introduction

·        Discussion about simplification of the well structure with cementing in the sentence “…Since 1998, some development wells have simplified their well structure and large sections of production casing have been produced with discontinuous cementing cement or uncured sections, resulting in very serious casing losses” is not clear. Please, clarify

·        Authors did not show what are the reasons of the poor electrochemical performance of the sacrificial anode materials. That is why the reader cannot understand why authors need improvement (the sentence “…. In order to improve the electrochemical performance of sacrificial anode materials, scholars have continuously added different elements to improve the electrochemical performance of aluminum zinc alloys”)

·        English is not proper in the sentence “…For example, in a high-temperature seawater environment above 60 , the potential reversal of the zinc anode relative to the protected steel material will occur, and the steel material will no longer be protected, but will accelerate corrosion [14]” . It is impossible to understand.

·        Authors need to explain in detail the authors’ approach of improving the casing corrosion resistance by “… a new type of zinc alloy sacrificial anode material which will be formed by strictly controlling the content of Al elements and adding Mn and Mg, etc.”

2.       Experimental

·        Authors did not show the materials’ chemical composition.

·        To show the dimensions of the sleeve in Figure 1. zinc alloy sacrificial anode sleeve.

·        Drawings of all samples listed in Table1(“ Two sets of galvanic corrosion specimens: (i) TP140 specimens protected by sacrificial anodes (3 parallel specimens); (ii) sacrificial anode specimens with TP140 attached (3 parallel specimens). Two sets of corrosion tests: (i) sacrificial anode specimens without casing attached (3 parallel specimens); (ii) TP140 specimens without sacrificial anode attached (3 parallel specimens)” need to be shown on Fig.2

3.      Results and Discussion

·        There is no analysis of experimental data shown in Fig.3. The information about corrosion rate presented in p.3.1. looks like a report, not paper.

·        P. 3.2.1. Macroscopic morphology cannot be understood from a metallography point of view. What is the scale of the images (Fig.4,5)?

·        The authors only demonstrated the images without any explanation

·         The data of p. 3.2.2. Microscopic morphology are not explained.

·        There is no corrosion reactions and phases description

·        Authors did not suggest mechanisms of sacrificial corrosion protection at high temperatures

·        Authors did not present a conclusion about the sacrificial materials’ effectiveness whilst declaring in the introduction that “…a new type of zinc alloy sacrificial anode material is formed by strictly controlling the content of Al elements and adding Mn and Mg, etc.” 

English is not proper

Reviewer 3 Report

The main comment to the article is that despite the detailed presentation of the results obtained, there is no analysis and discussion of them. The advantages or disadvantages of the studied anode material remain unclear, as well as the reasons for the changes occurring with it depending on temperature.

The description of the design of the working electrode is not entirely clear: «the working electrode was a stainless-steel specimen spot welded to a working electrode»

The polarization curves for the sacrificial anode in Figure 9 do not correspond to the description in the table and in the text. Obviously, in the figure you need to swap the curves obtained at 110°C and 120°C.

Author Response

Manuscript with a title: “Corrosion studies of temperature-resistant zinc alloy sacrificial anodes and casing pipe at different temperatures”

Number: materials-2453392

Dear Professor:

Thank you for your letter and the reviewer's comments on our manuscript. These comments are all valuable and very helpful in revising and improving our paper, as well as providing important guidance for our research. We have carefully studied the comments and have made corrections that we hope will meet with approval. The changes are highlighted in red in the revised manuscript. The point-by-point responses to the reviewers' and editor's comments are as follows:

Reviewer #3: The main comment to the article is that despite the detailed presentation of the results obtained, there is no analysis and discussion of them. The advantages or disadvantages of the studied anode material remain unclear, as well as the reasons for the changes occurring with it depending on temperature.

ResponseThank you for the reviewer’s comments of the manuscript. Thank you very much for your comments, and we have carefully revised the manuscript. We have added XRD related data, redescribed the electrochemical test results, and touched up the language throughout the text.

1) The description of the design of the working electrode is not entirely clear: «the working electrode was a stainless-steel specimen spot welded to a working electrode»

Response: Thank you for the review’s point of view. We describe electrochemical testing in detail in Section 2.2.

2) The polarization curves for the sacrificial anode in Figure 9 do not correspond to the description in the table and in the text. Obviously, in the figure you need to swap the curves obtained at 110°C and 120°C.

Response: Thank you for the review’s point of view. We have modified the data in Figure 11(Original Figure 9)and Table 3.

Best wishes

Yours sincerely

Wenhao Feng

Aug 15, 2023

PS: If the answer was not agreeable to the reviewer, we will try our best to find the reasonable explanation again.

Reviewer 4 Report

The manuscript is devoted to the study of a new type of zinc alloy sacrificial anode material simulating the high-temperature working conditions of a deep well in an oil field.  The manuscript is of great theoretical and practical interest, however there are some remarks.

1) The composition of the sacrificial anode material is not specified in the manuscript.

2) It is advisable to indicate in the manuscript the values of the protective effectiveness of sacrificial anode in relation to TP 140 under the studied conditions and present these data in the Abstract and Conclusion.

3) The last sentence in the abstract "The self-corrosion potential in the sacrificial anode material gradually increased with the in-crease of temperature, and the potential difference with TP140 gradually decreased." does not match Figure 9.

4) Authors should indicate in Table 2 which values are given in each column.

5) The data in Table 2 do not correspond to Fig. 9 for the sacrificial anode materials, since in Fig. 9, self-corrosion potential at 110° is higher than at 120o, and in Table 2 it is vice versa.

6) The authors should explain why the corrosion rate of TP140, both unprotected and protected, was greatest at 110° and decreased at 120°.

7) The authors should give recommendations in the Conclusion on the optimal conditions for the use of the sacrificial anodes under study for the protection of TP 140.

Author Response

Manuscript with a title: “Corrosion studies of temperature-resistant zinc alloy sacrificial anodes and casing pipe at different temperatures”

Number: materials-2453392

Dear Professor:

Thank you for your letter and the reviewer's comments on our manuscript. These comments are all valuable and very helpful in revising and improving our paper, as well as providing important guidance for our research. We have carefully studied the comments and have made corrections that we hope will meet with approval. The changes are highlighted in red in the revised manuscript. The point-by-point responses to the reviewers' and editor's comments are as follows:

Reviewer #4: The manuscript is devoted to the study of a new type of zinc alloy sacrificial anode material simulating the high-temperature working conditions of a deep well in an oil field.  The manuscript is of great theoretical and practical interest, however there are some remarks:

Response Thank you for the reviewers' comments on the manuscript. We have corrected the errors and problems as suggested, and the changes are highlighted in red in the revised manuscript.

1) The composition of the sacrificial anode material is not specified in the manuscript.

Response: Thank you for the review’s point of view. The chemical composition of the sacrificial anode material is given in section 2.1 of the manuscript.

2) It is advisable to indicate in the manuscript the values of the protective effectiveness of sacrificial anode in relation to TP 140 under the studied conditions and present these data in the Abstract and Conclusion.

Response: Thank you for the review’s point of view. In section 3.1, we add the protection efficiency of sacrificial anodes at different temperatures. We mention in the abstract the values of the protective effect of sacrificial anodes relative to TP 140 under the conditions studied and add in the conclusions that the corrosion rate of TP 140 protected by sacrificial anodes at 100-120°C is only one tenth of that under unprotected conditions.
3) The last sentence in the abstract "The self-corrosion potential in the sacrificial anode material gradually increased with the in-crease of temperature, and the potential difference with TP140 gradually decreased." does not match Figure 9.

Response: Thank you for the review’s point of view. We have modified the presentation in the abstract.

 4) Authors should indicate in Table 2 which values are given in each column.

Response: Thank you for the review’s point of view. We have made the relevant changes.
5) The data in Table 2 do not correspond to Fig. 9 for the sacrificial anode materials, since in Fig. 9, self-corrosion potential at 110° is higher than at 120o, and in Table 2 it is vice versa.

Response: Thank you for the review’s point of view. We have corrected the data in Table 3(Original table 2), which was confusing due to our inadvertence.

6) The authors should explain why the corrosion rate of TP140, both unprotected and protected, was greatest at 110° and decreased at 120°.

Response: Thank you for the review’s point of view. The issue is explained in section 3.1 and highlighted in red.

7) The authors should give recommendations in the Conclusion on the optimal conditions for the use of the sacrificial anodes under study for the protection of TP 140.

Response: Thank you for the review’s point of view. Recommendations are provided in the conclusions.

Best wishes

Yours sincerely

Wenhao Feng

Aug 15, 2023

PS: If the answer was not agreeable to the reviewer, we will try our best to find the reasonable explanation again.

Reviewer 5 Report

This manuscript explores a new type of high temperature resistant zinc alloy sacrificial anode material and TP140 casing by simulating the high temperature working conditions. The author’s new anode is made of Zn-Al-Cd alloy with controlling appropriate amounts of Al, Mn, and Mg. The author analyzed the TP140 morphology with and without sacrificial anode by ultra-deep filed microscopy, SEM, and EDS under different temperature at 100, 110, and 120. Also, the electrochemical tests were conducted by three electrode system. This manuscript shows the new sacrificial anode can protect the corrosion of TP140, however, some parts need to be addressed being considered for publication. Therefore, I recommend the manuscript to be published after minor revisions.

Comment 1:

This paper does not flow smoothly; therefore, it should be proofread in English.

Comment 2:

In the experimental section, the method of manufacture new type of sacrificial anode material and the substance content of the constituent materials should be described.

Comment 3:

All results in this paper show that anodes with and without TP140 casing have the highest corrosion rate at 110 ℃, followed by 120 ℃, and 100 ℃. However, the authors do not explain the mechanism by which corrosion increases and decreases with temperature, and the amount of data is insufficient to support their claims.

Comment 4:

In figure 3, the author argues that unprotected TP 140 shows a high corrosion rate at every temperature. However, the claims in figure 5 doesn’t make sense that no significant corrosion cracking were observed on the surface of both sacrificial anodes. The author should explain the reason.

Comment 5:

In the EDS analysis part, the author explains a higher content of Ca in the products are unprotected samples. However, the absence of the manufacturing process of TP140/sacrificial anode material, there is no way to understand how well the sacrificial anode material distributed on the TP140 surface. So that the EDS analysis image seems to be taken at too high magnification. The EDS mapping is recommended to be taken at whole size of image in figure 6, 7, 8.

Author Response

Manuscript with a title: “Corrosion studies of temperature-resistant zinc alloy sacrificial anodes and casing pipe at different temperatures”

Number: materials-2453392

Dear Professor:

Thank you for your letter and the reviewer's comments on our manuscript. These comments are all valuable and very helpful in revising and improving our paper, as well as providing important guidance for our research. We have carefully studied the comments and have made corrections that we hope will meet with approval. The changes are highlighted in red in the revised manuscript. The point-by-point responses to the reviewers' and editor's comments are as follows:

Reviewer #5: This manuscript explores a new type of high temperature resistant zinc alloy sacrificial anode material and TP140 casing by simulating the high temperature working conditions. The author’s new anode is made of Zn-Al-Cd alloy with controlling appropriate amounts of Al, Mn, and Mg. The author analyzed the TP140 morphology with and without sacrificial anode by ultra-deep filed microscopy, SEM, and EDS under different temperature at 100, 110, and 120. Also, the electrochemical tests were conducted by three electrode system. This manuscript shows the new sacrificial anode can protect the corrosion of TP140, however, some parts need to be addressed being considered for publication. Therefore, I recommend the manuscript to be published after minor revisions.

Response Thank you for the reviewers' comments on the manuscript. We have corrected the errors and problems as suggested, and the changes are highlighted in red in the revised manuscript.

1) This paper does not flow smoothly; therefore, it should be proofread in English.

Response: Thank you for the review’s point of view. Extensive changes have been made to the English presentation.

2) In the experimental section, the method of manufacture new type of sacrificial anode material and the substance content of the constituent materials should be described.

Response: Thank you for the review’s point of view. The chemical composition of the sacrificial anode material is given in section 2.1 of the manuscript.
3) All results in this paper show that anodes with and without TP140 casing have the highest corrosion rate at 110
, followed by 120 , and 100 . However, the authors do not explain the mechanism by which corrosion increases and decreases with temperature, and the amount of data is insufficient to support their claims.

Response: Thank you for the review’s point of view. In section 3.1, we add the protection efficiency of sacrificial anodes at different temperatures. And it explains that TP140 corrodes the fastest at 110℃.

4) In figure 3, the author argues that unprotected TP 140 shows a high corrosion rate at every temperature. However, the claims in figure 5 doesn’t make sense that no significant corrosion cracking were observed on the surface of both sacrificial anodes. The author should explain the reason.

Response: Thank you for the review’s point of view. We have supplemented the macroscopic morphology of TP140 and sacrificial anode materials after corrosion at different temperatures with specific representations.
5) In the EDS analysis part, the author explains a higher content of Ca in the products are unprotected samples. However, the absence of the manufacturing process of TP140/sacrificial anode material, there is no way to understand how well the sacrificial anode material distributed on the TP140 surface. So that the EDS analysis image seems to be taken at too high magnification. The EDS mapping is recommended to be taken at whole size of image in figure 6, 7, 8.

Response: Thank you for the review’s point of view. In order to further analyze the corrosion products on the surface of TP140, we added the XRD results of the corrosion products of the protected and unprotected TP140 samples at 100°C. The XRD data can be better combined with the EDS results to understand the products on the surface of the samples.

Best wishes

Yours sincerely

Wenhao Feng

Aug 15, 2023

PS: If the answer was not agreeable to the reviewer, we will try our best to find the reasonable explanation again.

Round 2

Reviewer 1 Report

The authors answered some of my previous comments. The paper has been improved. However, few issues remained unanswered. The following points should be clarified before accepting the paper for publication:

1.It is advised to number all lines in the manuscript to facilitate orientation.

2.You haven’t specified the concentration of cadmium in the Zn-Al-Cd alloy (see the end of first paragraph in materials section).

3.It is not clear if the metal percents are atomic or weight. Specify it in the materials section and Table 1.

4.It is recommended to provide a photograph of the galvanic corrosion test, i.e., the sacrificial anode – steel couple and its placement in a special hanger for the autoclave. You should provide a schematic for the corrosion experiment as it was quite complex (the reference electrode was located outside of the autoclave, see the red text on p. 3). It will assist the readers in comprehension of your paper.

5.The equation p. 5 should be numbered properly, i.e., (1) instead of (2.1).

6.There is no scale bar in Fig. 4.

7.There should be a legend in the upper column of Table 3. I presume the left values correspond to Ecorr and the right values to jcorr. Specify them as such in the table.

8.I don’t see any results for galvanic cell experiments except weight loss results (Fig. 3). Tafel curves are given for separated anode and steel only (Fig. 11). If you measured the galvanic couple (steel-sacrificial anode) in a three-electrode cell, as specified in Experimental part (the red text on p.3), you should present the results too.

9.Have you also measured open circuit potentials? If yes, how do they compare with Ecorr from Tafel curves?

Author Response

Manuscript with a title: “Corrosion studies of temperature-resistant zinc alloy sacrificial anodes and casing pipe at different temperatures”

Number: materials-2453392

Dear Professor:

Thank you for your letter and the reviewer's comments on our manuscript. These comments are all valuable and very helpful in revising and improving our paper, as well as providing important guidance for our research. We have carefully studied the comments and have made corrections that we hope will meet with approval. The changes are highlighted in red in the revised manuscript. The point-by-point responses to the reviewers' and editor's comments are as follows:

Reviewer #2: 1.It is advised to number all lines in the manuscript to facilitate orientation.

ResponseThank you for your comments on the manuscript. We have numbered all lines in the manuscript to facilitate orientation.

  1. You haven’t specified the concentration of cadmium in the Zn-Al-Cd alloy (see the end of first paragraph in materials section).

ResponseThank you for the review’s point of view. We have already indicated the concentration of cadmium in Zn-Al-Cd alloys in section 2.1 and highlighted it in yellow.

  1. It is not clear if the metal percents are atomic or weight. Specify it in the materials section and Table 1.

ResponseThank you for the review’s point of view. We modify Table 1 as follows “Elemental composition of TP140 (wt.%)”.

  1. It is recommended to provide a photograph of the galvanic corrosion test, i.e., the sacrificial anode – steel couple and its placement in a special hanger for the autoclave. You should provide a schematic for the corrosion experiment as it was quite complex (the reference electrode was located outside of the autoclave, see the red text on p. 3). It will assist the readers in comprehension of your paper.

ResponseThank you for the review’s point of view. We have added a schematic of the electrochemical test.

  1. The equation p. 5 should be numbered properly, i.e., (1) instead of (2.1).

ResponseThank you for the review’s point of view. We have changed the labelling of the equation on page 5.

  1. There is no scale bar in Fig. 4.

ResponseThank you for the review’s point of view. We have modified Figure 4 by adding the scale.

  1. There should be a legend in the upper column of Table 3. I presume the left values correspond to Ecorr and the right values to jcorr. Specify them as such in the table.

ResponseThank you for the review’s point of view. We have modified Table 3.

  1. I don’t see any results for galvanic cell experiments except weight loss results (Fig. 3). Tafel curves are given for separated anode and steel only (Fig. 11). If you measured the galvanic couple (steel-sacrificial anode) in a three-electrode cell, as specified in Experimental part (the red text on p.3), you should present the results too.

ResponseThank you for the review’s point of view. We didn’t measured the galvanic couple (steel-sacrificial anode) in a three-electrode cell.

  1. Have you also measured open circuit potentials? If yes, how do they compare with Ecorr from Tafel curves?

ResponseThank you for the review’s point of view. Experiments are carried out when the open circuit potential is stable, unfortunately, we didn't measure the open-circuit potential.

Best wishes

Yours sincerely

Wenhao Feng

Aug 22, 2023

PS: If the answer was not agreeable to the reviewer, we will try our best to find the reasonable explanation again.

Reviewer 2 Report

N/A

Author Response

Thank you for your comments on the manuscript again.

Reviewer 3 Report

The article has been substantially updated. It seems to me that the conclusions can be improved. For example, paragraphs 1 and 2 repeat the experimental part. It would be better to make them generalizing, noting the specific advantages or disadvantages of the material.

Author Response

Thank you for your comments on the manuscript. The changes are highlighted in yellow in the text, and we have followed your suggestion to make the conclusion more concise.